# Estimating word co-occurrence probabilities from pretrained static embeddings using a log-bilinear model

**Richard Futrell**
Department of Language Science
University of California, Irvine
`rfutrell@uci.edu`

## Abstract

I investigate how to use pretrained static word embeddings to deliver improved estimates of bilexical co-occurrence probabilities: conditional probabilities of one word given a single other word in a specific relationship. Such probabilities play important roles in psycholinguistics, corpus linguistics, and usage-based cognitive modeling of language more generally. I propose a log-bilinear model taking pretrained vector representations of the two words as input, enabling generalization based on the distributional information contained in both vectors. I show that this model outperforms baselines in estimating probabilities of adjectives given nouns that they attributively modify, and probabilities of nominal direct objects given their head verbs, given limited training data in Arabic, English, Korean, and Spanish.

## 1 Introduction

Word co-occurrence probabilities are a key ingredient in usage-based cognitive models of language. By word co-occurrence probabilities, I mean the probability of a word $w$ given some other single word $c$, $p(w \mid c)$, where words $w$ and $c$ have some specific relationship, for example adjectives that attributively modify nouns or nouns serving as direct objects of verbs (Gries and Durrant, 2020).

These co-occurrence probabilities are psycholinguistically relevant because they feed into information-theoretic measures of 'thematic fit' and selectional restriction (Resnik, 1996; Lapata et al., 1999; Padó et al., 2007; Vecchi et al., 2017) which are relevant in predicting human online processing difficulty (e.g. McRae et al., 1998; Trueswell et al., 1994), and play a key role in language acquisition (Erickson and Thiessen, 2015). Most prominently, the widely-used pointwise mutual information (PMI) measure of association strength, $\mathrm{PMI}(w, c) = \log \frac{p(w|c)}{p(w)}$ (Fano, 1961; Church and Hanks, 1990), relies on these condi-

tional probabilities as an input. PMI makes appearances in models of grammar induction from text (Magerman and Marcus, 1990; Yuret, 1998; Clark and Fijalkow, 2020; Hoover et al., 2021), online sentence comprehension and production (Futrell et al., 2020b; Ranjan et al., 2022), and quantitative theories of word order variation (Futrell et al., 2020a; Sharma et al., 2020).

Word co-occurrence probabilities are hard to estimate accurately from text data because empirical counts of a particular pair of words in a particular relation are often sparse. This limitation makes it hard to evaluate cognitive theories that operate on co-occurrence probabilities. Although high-performance pretrained language models now exist (Radford et al., 2019; Devlin et al., 2019, etc.), the probabilities of interest often cannot be read off of these models directly, because $w$ and $c$ might be defined by relations that cannot be straightforwardly detected in terms of linear word order or templates. For example, suppose we are interested in the distribution of adjectives attributively modifying a noun in English. It would not do to ask a language model for the distribution of words immediately preceding a noun, because some of these words will not be attributive adjectives.

I propose to improve the estimation of word co-occurrence probabilities by leveraging pretrained static word embeddings to enhance generalization from potentially small training sets. My method enables generalization based on the semantic and syntactic information contained in word embeddings for both words $w$ and $c$.

## 2 Model

**Setting** We are given a **vocabulary** of words $V$, a finite **target word set** $W \subseteq V$, a dataset of $N$ pairs of words $\{\langle w_i, c_i \rangle\}_{i=1}^{N}$ where the **target word** $w$ is an element of target word set $W$ and the **context word** $c$ is an element of the full vocabulary $V$, and a pretrained mapping from words to

$D$-dimensional static embeddings $E : V \to \mathbb{R}^D$. Supposing the dataset consists of iid samples from some distribution $p(w, c) = p(c)p(w \mid c)$, our goal is to find a conditional distribution $q(w \mid c)$ with support $W$ to approximate $p(w \mid c)$ in a way that leverages the static embeddings $E$.

**Proposed model**   I propose a log-bilinear model (Mnih and Hinton, 2007, 2008) using word embeddings as input:[1]

$$q(w \mid c) = \frac{1}{Z(c)} \exp\left\{\phi(\mathbf{w})^\top \mathbf{A} \psi(\mathbf{c})\right\} \quad (1)$$

$$Z(c) = \sum_{w \in W} \exp\left\{\phi(\mathbf{w})^\top \mathbf{A} \psi(\mathbf{c})\right\}, \quad (2)$$

where $\mathbf{w} = E(w)$ and $\mathbf{c} = E(c)$ are the static embeddings of target word $w$ and context word $c$ respectively, the **target word encoder** $\phi(\cdot) : \mathbb{R}^D \to \mathbb{R}^K$ and **context word encoder** $\psi(\cdot) : \mathbb{R}^D \to \mathbb{R}^L$ are functions which may be parameterized as feed-forward neural networks with parameters denoted $\phi$ and $\psi$ respectively, and $\mathbf{A}$ is a $K \times L$ interaction matrix. The model parameters $\phi$, $\psi$, and $\mathbf{A}$ are trained to minimize the cross-entropy loss

$$J(\phi, \psi, \mathbf{A}) = -\sum_{n=1}^{N} \log q(w_n \mid c_n). \quad (3)$$

**Modeling decisions**   A modeler applying this approach needs to make a number of decisions, including the choice of static word embeddings and the structure of the word encoders $\phi(\cdot)$ and $\psi(\cdot)$. It is also possible to set $\psi = \phi$, using the same function to encode both the target word and the context word; this setup can reduce the number of parameters at the cost of less flexibility in fitting the training data.

Another major modeling decision involves the target word vocabulary $W$, which determines the support of $q(w \mid c)$ and is summed over during the calculation of the partition function (Eq. 2). In some cases, the modeler may not have access to a finite set $W$ of possible target words. As long as the full vocabulary $V$ is finite, it is possible to set $W = V$ and learn a probability distribution with support on all words in $V$.

Setting $W = V$ has the advantage that it allows the modeler not to commit to any particular target word set, thus avoiding the risk of prematurely excluding legitimate target words. It has the

---

[1] I have suppressed bias terms from the notation.

disadvantages that (1) calculation of the partition function (Eq. 2) is slower and/or more memory intensive, and (2) the learning problem is more difficult because probability mass is initially spread over the set $V$ as opposed to a potentially much smaller set $W$.

**Implementation**   In all experiments reported below, stochastic gradient descent is performed using the Adam algorithm with default initial learning rate (Kingma and Ba, 2015). All experiments are implemented in PyTorch with use of `opt_einsum` to compute the partition function (Smith and Gray, 2018; Paszke et al., 2019).

To handle out-of-vocabulary items, I include an unknown-word symbol `UNK` in the target word set $W$ and full vocabulary $V$. If a target word $w$ in a dataset is not present in the target word set $W$, or a context word $c$ is not present in the full vocabulary $V$, then that word is mapped to `UNK`. In the embedding map, `UNK` is assigned to a normalized random vector drawn from a Gaussian distribution.

## 3   Related work

Distributional similarity information has been used to improve modeling of word co-occurrence probabilities in previous work. Dagan et al. (1994, 1999) defined a kernel-based interpolated language model where probability mass is explicitly spread over similar words, with variant models along these lines found in Wang et al. (2005) and Yarlett (2008). These models leverage similarity information about target words but not context words. In contrast, Bíró et al. (2007) proposed a method which uses similarity information about the context word but not the target word. Toutanova et al. (2004) developed a method that can exploit similarity information about both target and context, using a Markov Chain algorithm incorporating distributional and WordNet similarities. None of this previous work derived word similarity information from pretrained embeddings, because such embeddings did not exist at the time.

The log-bilinear model for conditional word probabilities was introduced in a language modeling context by Mnih and Hinton (2007, 2008). Mikolov et al. (2013a) influentially proposed to use the vector representations output by the word encoder in such a model as general word embeddings. The current work aims to return log-bilinear models to their language modeling roots, evaluating the capabilities of these models to estimate co-occurrence

probabilities using pretrained embeddings as input, with a focus on word distributions where training data is limited. Here the target word vocabulary is typically small enough that the partition function (Eq. 2) can be computed directly on modern hardware, so that approximations such as noise-contrastive estimation (Mikolov et al., 2013b) are not necessary.

Recently Nikkarinen et al. (2021) introduced a neural-Bayesian nonparametric estimator for probability distributions on single words. Their setting has an unknown and generally infinite vocabulary $V$, and their model generalizes using a character-level LSTM. In contrast, the current model assumes a pre-existing known vocabulary $V$ with embeddings, and generalizes based on those embeddings. A hybrid model may be possible in future work.

A related literature in corpus linguistics and NLP has explored the nature of restricted binary word co-occurrences, called collocations (for recent examples, see Savary et al., 2017; Kutuzov et al., 2017; Garcia et al., 2021; Espinosa Anke et al., 2021). This work focuses narrowly on the estimation of bilexical conditional probabilities, which are often inputs to models for collocation detection.

## 4 Experiments

I study the ability of the embedding-based log-bilinear model to estimate conditional distributions for (1) adjectives attributively modifying nouns and (2) nominal direct objects modifying verbs, in Arabic, English, Korean, and Spanish. I compare the model against baselines:

- Additive smoothing with $\alpha = 1$:

$$p_{\text{add}}(w \mid c; \alpha) \propto \text{count}(c, w) + \alpha,$$

  where $\text{count}(c, w)$ is the frequency of the pair of words $c$ and $w$ in the training data.

- An interpolated smoothed estimator:

$$p_{\text{interp}}(w \mid c) = p_{\text{add}}(w \mid c; \alpha) + \lambda p_{\text{MLE}}(w),$$

  where $p_{\text{MLE}}$ is a maximum likelihood estimate, $\lambda = \frac{1}{4}$, and $\alpha = 1$.

- A softmax distribution on target words as a function of the context word embedding $\mathbf{c}$ (as proposed by Bíró et al., 2007):

$$p_{\text{softmax}}(w \mid c) \propto \exp\left\{\theta_w^\top \psi\left(\mathbf{c}\right)\right\},$$

where $\theta_w$ is an optimized weight vector for the target word $w$. This baseline uses the context word embedding $\mathbf{c}$ but not the target word embedding $\mathbf{w}$. It is equivalent to having the target word encoder return a one-hot vector representation of target word $w$.

- Models without word encoders, achieved by setting $\phi(\cdot)$ and $\psi(\cdot)$ to identity functions. Such models decode target words from the word embeddings directly.

All baselines are subject to the same vocabulary restrictions and out-of-vocabulary policy as the full log-bilinear models. As a standard test metric, I report the average negative log likelihood (NLL) of held-out data. I report NLLs for the full test set, as well as the challenging subset of the test set consisting of word pairs where the context word was never seen during training.

Below, I describe the experimental setting for the two tasks, and then I describe the results.

### 4.1 Distribution of attributive adjectives given nouns

I examine the distribution of attributive adjectives given the nouns that they modify, for example adjectives like *red* modifying nouns like *ball* in phrases like *the red ball*.

**Data**  I use Universal Dependencies (UD) 2.8[2] (Nivre et al., 2020) and the automatically-parsed Wikipedia datasets released as part of the CoNLL 2017 Shared Task (Zeman et al., 2017) as a source of attributive adjective–noun pairs. I extract all pairs of words linked by a dependency of type *amod* where the head has universal part-of-speech (UPOS) NOUN and the dependent has UPOS ADJ. I represent the pair using the downcased wordforms of the adjective and noun.

For each language, I use the fastText aligned word vectors (Bojanowski et al., 2017; Joulin et al., 2018),[3] limiting the vocabulary set $V$ to the top 200,000 vectors by frequency. For the target word vocabulary $W$, I take the 10,000 most frequent wordforms among all attributive adjectives extracted from the entire CoNLL Wikipedia dataset.

As training sets, I use 99,000 adjective–noun pairs drawn randomly from the Wikipedia datasets for each language, so training set size is fixed

[2] http://hdl.handle.net/11234/1-3687
[3] https://fasttext.cc/docs/en/aligned-vectors.html

| Data | Attributive adjectives given nouns | | | | | | Direct objects given verbs | | | | | |
| | | | Softmax | | Log-Bilinear | | | | Softmax | | Log-Bilinear | |
| | Add. | Interp. | No Enc. | Enc. | No Enc. | Enc. | Add. | Interp. | No Enc. | Enc. | No Enc. | Enc. |
|---|---|---|---|---|---|---|---|---|---|---|---|---|
| Arabic | 8.50 | 7.05 | 8.31 | 8.04 | **5.79** | 5.89 | 9.78 | 9.78 | 9.17 | 9.00 | 8.63 | **8.47** |
| *Unseen c* | 9.15 | 9.60 | 8.31 | 8.52 | **6.93** | 6.98 | 9.71 | 9.84 | 9.03 | 8.86 | 9.09 | **8.76** |
| English | 8.75 | 7.17 | 7.15 | 7.16 | **6.40** | 6.41 | 9.64 | 8.99 | 8.64 | 8.58 | 8.16 | **8.04** |
| *Unseen c* | 9.01 | 8.40 | 7.21 | 7.22 | 6.99 | **6.96** | 9.89 | 9.96 | 8.62 | 8.56 | 8.39 | **8.35** |
| Spanish | 8.70 | 7.49 | 8.13 | 8.10 | **6.27** | **6.27** | 9.70 | 9.10 | 8.64 | 8.52 | 7.96 | **7.84** |
| *Unseen c* | 9.17 | 9.50 | 8.15 | 8.21 | 7.16 | **7.09** | 9.80 | 9.62 | 8.48 | 8.48 | 8.35 | **8.18** |
| Korean | 7.96 | 5.39 | 5.51 | 5.61 | **4.81** | 4.82 | 9.71 | 9.76 | 9.20 | 9.18 | 8.34 | **7.99** |
| *Unseen c* | 7.16 | 5.92 | 5.45 | 5.48 | 5.44 | **5.40** | 9.67 | 9.91 | 9.16 | 9.14 | 9.58 | **8.76** |

Table 1: Average NLLs of adjectives given nouns and direct objects given verbs in UD corpora for models and baselines. 'Add.' is the additive smoothing baseline. 'Enc.' and 'No Enc.' refer to models with and without word encoders, respectively. *Unseen c* indicates performance on pairs where the context (the head noun for adjectives given nouns, and the head verb for direct objects given verbs) was never observed at train time.

across languages. I use an additional 1,000 pairs from the Wikipedia datasets as development sets for hyperparameter tuning and early stopping, and for test sets I extract all pairs from the relevant UD corpora.[4] Pairs where the target word $w$ is not in the target word vocabulary $W$ are removed from the development and test sets.

**Training and hyperparameters**   Each model is trained for the number of iterations that gives minimum loss on the Wikipedia dev set. The word encoders are feed-forward neural networks with one hidden layer of 300 units and an output layer of 300 units, with ReLU activation. In training, I use batch size 32; I also experimented with batch size 512 but this resulted in rapid overfitting.

### 4.2   Distribution of nominal direct objects given verbs

I examine the distribution of nominal direct objects given verbs; for example, from a sentence such as *I kicked the red ball*, one would be interested in the probability of the direct object *ball* given its head noun *kicked*. All procedures here are the same as for the distribution of attributive adjectives given nouns except as described below.

**Data**   I extracted direct objects as all pairs of words linked in a dependency of type *obj* where the head has UPOS VERB and the dependent has UPOS NOUN. Because nouns are more open-class than adjectives, I used a target word vocabulary of size 20,000.

---

[4]For English, I concatenate EWT and GUM. For Arabic, I concatenate NYUAD and PADT. For Spanish, I concatenate AnCora and GSD. For Korean, I concatenate Kaist and GSD.

### 4.3   Results

Results are shown in Table 1. The log-bilinear models outperform all others. In several cases (see for example Spanish and Korean adjectives), only the log-bilinear model is capable of outperforming the interpolated baseline.

When predicting adjectives from nouns, the log-bilinear models without word encoders sometimes outperform those with word encoders. These is perhaps not surprising: the input word embeddings were trained to be used in a log-bilinear skip-gram probability model, so they already form useful representations for word prediction.

Overall performance on predicting objects from verbs is worse than when predicting adjectives from nouns. This reflects the harder nature of the task and the larger support size required to model nouns rather than adjectives.

### 4.4   Additional experiments

I also trained full log-bilinear models with a number of other settings. I found that tying the word and context encoders does not substantially change performance, but that fine-tuning the input word embeddings leads to severe overfitting. Removing the target word vocabulary restriction (setting $W = V$) also substantially negatively impacts performance: for adjectives, the best test set NLL is 6.57 for Arabic, 6.75 for English, 6.95 for Spanish, and 4.89 for Korean.

### 5   Conclusion

I evaluated log-bilinear modeling as means to leverage pretrained word embeddings for the es-

timation of co-occurrence probabilities in different syntactic configurations. I found that this method delivers accurate probability estimates across languages, outperforming baselines. This method will be useful in all applications requiring such probabilities. Code implementing the method can be found at `https://github.com/langprocgroup/vectorprob`.

## Acknowledgments

This work was supported by NSF Grant #1947307 and an NVIDIA GPU Grant to the author. I thank Charles Torres, Gregory Scontras, and William Dyer for helpful discussion.

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
