# OpenReview forum: "Estimating word co-occurrence probabilities from pretrained static embeddings using a log-bilinear model"
_aclweb.org/ACL/2022/Workshop/CMCL — CMCL 2022_

### Official Review · Reviewer_JehU · 2022-03-22
**An old model made new with word embeddings**

**Rating:** 7
**Confidence:** 4

**Review:**

This paper proposes a way to estimate word co-occurrence probabilities by leveraging word embeddings. The method uses a log-linear bilinear model that had been proposed by Mnih and Hinton some time ago, but marries it with pre-trained word embeddings that are derived from a much larger corpus, and which are not specialized to specific types of word co-occurrence. The results are fairly impressive: the trained model yields lower negative log likelihood in all cases, as compared to a number of other baselines. Overall, I found the approach sensible, if not groundbreaking, and the evaluation reasonably compelling.  I have a couple of points I'd like to raise:

1. There is considerable variation in performance across the languages (In the adj-noun case, Korean is uniformly the best performing, and . Arabic and Spanish the worst, though there is some variation across models. Things are different in the verb-object case.) Does this stem from the differently-sized test sets (I am assuming that the training data were identically sized across languages), or from differences in the fasttext embeddings (perhaps stemming from differences in training data again), or from differences in the languages, or something else? It would be helpful for the paper to say something about this. For one thing, it would be useful to give sizes of the datasets, as well as measures of the number of distinct context and target words.

2. The paper starts by talking about the cognitive utility of having estimates of co-occurrence probability, and the difficulty of doing that in the context of limited (parsed) data. It would be helpful to know the degree to which the proposed method is doing better in contexts of limited data.  In particular, it would be interesting to know how performance of the different models varies as the training data changes in size. For models that can do better with less data, we'd expect to see convergence with smaller datasets.  It would also be useful to know just how well the different models can do in the limit, if we give them limitless data (or nearly as much as we have). Part of the lower performance might be due to difficulty in parameter estimation, and part might be due to the structure of the model, so it'd be useful to know the degree to which the proposed method is really solving the sparse data problem.

3. Given that this is for CMCL, I'd like to have seen some discussion about the cognitive plausibility of the approach. it is commonly thought that word embeddings represent lexical semantics, but clearly similar meaning will not be sufficient to capture co-occurence probability (as in cases of differences in register, or word collocations). Instead, the important property of word embeddings that is being exploited here is that they are derived exactly from attempts to predict co-occurrence. From that perspective, should we expect that any specific way of deriving word embeddings would perform better than another?

Specific comments:

line 61: why can't we use a language model to get predicted adjectives by renormalizing the predicted distribution to limit the support to the same 20k adjectives you look at here? How do the distributions we get in this way from existing large language models (e.g., GPT-3 or BERT) compare to what we get here?

line 204: why these hyper parameters?

line 282: what does it mean that nouns are more open class? That there are more of them? If so, say that. They are both clearly open class.

---

### Official Review · Reviewer_tnxr · 2022-03-23
**A potentially useful tool for psycholinguists**

**Rating:** 8
**Confidence:** 4

**Review:**

This paper describes an estimation technique for word co-occurrence probabilities, unifying two relatively old pieces of technology: static word embeddings and a log-bilinear model from Mnih and Hinton 2008. The authors show that word pair probabilities remain useful in some psycholinguistic analyses and argue that large pretrained LMs which are now the state of the art in word prediction may not be able to read off the probabilities of interest without extensive computational effort (e.g. evaluating every word in the target set, in a particular context). Their model outperforms some basic LM smoothing techniques and is relatively lightweight and easy to estimate.

Both the mathematical framework and the presentation of related work are admirably clear given the limited space available. Results are very good across multiple languages (variants of this technique outperform very reasonable baselines, such as learning a softmax distribution over the target word, by large margins on many languages).

Given the space constraints, I would not expect to see a lot of task-specific analyses here. It would be nice to have an idea of how performance varies in training set size, and why performance is so low when the target vocabulary is unrestricted. These are likely to be important questions for psycholinguists considering how to use this technique.

---

### Official Review · Reviewer_8X5b · 2022-03-28
**Looking for word co-occurrence probabilities in pre-trained word embeddings**

**Rating:** 7
**Confidence:** 4

**Review:**

This paper recycles the log-bilinear model (LBL), in a clever way to estimate bilexical co-occurrence probabilities using word embeddings. The motivation behind this work is that estimating word pair probabilities is often difficult due to the sparse nature of this data. Yet, having accurate word pair probabilities is relevant when conducting psycholinguistic experiments that evaluate usage-based cognitive language models (also likely in neurolinguistics it could be used as a useful baseline when trying to understand/evaluate the performance of word embeddings in decoding neural language activity in the brain). The authors propose to re-use the log-bilinear model from Mnih and Hinton 2007, arguably the simplest of neural language models, to estimate word-pair probabilities, specifically, by using as input to the LBL pre-trained static word embeddings (fastText word-vectors).

The performance of the LBL model is examined across four different languages (Arabic, English, Spanish, and Korean) while controlling for the size of the training data. They specifically look at predicting the distribution of attributive adjectives given nouns and nominal direct objects given verbs. The results are convincing with the log-bilinear model outperforming several baselines (i.e., lower negative log
likelihood). Particularly revealing is that it also consistently did better than a softmax distribution on target words as a function of the context word embedding (or when only using a word embedding for the context not the target word) confirming the utility of using word embeddings. I imagine that due to the light-weight nature of this model it would be of particular interest to language
researchers in cognitive science. The authors are also planning on making the code (trained model) available after publication. The only issue I have is that the paper doesn’t really address the sparse nature problem they originally used to partially motivate this work. To this extent, it would be good to note if the authors observed any significant performance advantages when changing the size of the
training data set (compared to the baselines)? It would be good to give users a rough guideline on these metrics so that this model can be properly used (also performance using other common word embeddings other than fastText).

Typo: Page 4, Section 4.3 Results, “These is perhaps not surprising…” should be “This is perhaps not surprising”.

---

### Decision · Program_Chairs · 2022-03-29

Accept